# Psychological detachment from work predicts mental wellbeing of working-age adults: Findings from the 'Wellbeing of the Workforce' (WoW) prospective longitudinal cohort study

Holly Blake [1,2]*, Juliet Hassard[3], Louise Thomson [4,5], Wei Hoong Choo [4], Teixiera Dulal-Arthur[4], Maria Karanika-Murray[6], Lana Delic[4], Richard Pickford [7], Lou Rudkin[5]

1 School of Health Sciences, University of Nottingham, Nottingham, United Kingdom, 2 NIHR Nottingham Biomedical Research Centre, Nottingham, United Kingdom, 3 Queen's University Belfast, Belfast, Northern Ireland, 4 School of Medicine, University of Nottingham, Nottingham, United Kingdom, 5 Institute of Mental Health, Nottingham, United Kingdom, 6 School of Business, University of Leicester, Leicester, United Kingdom, 7 Nottingham Civic Exchange, Nottingham Trent University, Nottingham, United Kingdom

* holly.blake@nottingham.ac.uk

## Abstract

### Background

There is an urgent need to better understand the factors that predict mental wellbeing in vocationally active adults during globally turbulent times.

### Aim

To explore the relationship between psychological detachment from work (postulated as a key recovery activity from work) in the first national COVID-19 lockdown with health, wellbeing, and life satisfaction of working age-adults one year later, within the context of a global pandemic.

### Methods

Wellbeing of the Workforce (WoW) was a prospective longitudinal cohort study, with two waves of data collection (Time 1, April-June 2020: T1 n = 337; Time 2, March-April 2021: T2 = 169) corresponding with the first and third national COVID-19 lockdowns in the UK. Participants were >18 years, who were employed or self-employed and working in the UK. Descriptive and parametric (t-tests and linear regression) and nonparametric (chi square tests) inferential statistics were employed.

### Results

Risk for major depression (T1: 20.0% to T2: 29.0%, p = .002), poor general health (T1: 4.7% to T2: 0%, p = .002) and poor life satisfaction (T1: 15.4% to T2: 25.4%, p = .002) worsened over time, moderate-to-severe anxiety remained stable (T1: 26.1% to T2: 30.2%, p = .15). Low psychological detachment from work was more prevalent in the first wave (T1: 21.4%

**Data Availability Statement:** The dataset used in this study is available from the Nottingham Research Data Management Repository (http://doi.org/10.17639/nott.7435).

**Funding:** The study was supported by Nottingham Healthcare NHS Foundation Trust Institute of Mental Health, National Institute for Health Research (NIHR) Research Capability Funding (RCF), and the Mental Health and Productivity Pilot (https://mhpp.me) which is funded by Employers, Health and Inclusive Employment (EHIE). These organisations had no role in the design of the Wellbeing at Work (WoW) study; in the collection, analyses, or interpretation of data; in the writing of the manuscript, or in the decision to publish the results.

**Competing interests:** The authors have declared that no competing interests exist.

and T2: 16.0%), with a moderate improvement observed from T1 to T2 ($t$ (129) = -7.09, $p <$ .001). No differences were observed with work status (employed/self-employed), except for self-employed workers being more likely to report poor general health at T1 (16.1%, $p =$ .002). Better psychological wellbeing, lower anxiety and higher life satisfaction at T2 were observed in those who reported better psychological detachment from work at T1 (β = .21, $p$ = .01; β = -.43, $p <$ .001; β = .32, $p =$ .003, respectively), and in those who improved in this recovery activity from T1 to T2 (β = .36, $p <$ .001; β = -.27, $p <$ .001; β = .27, $p =$ .008, respectively), controlling for age, gender and ethnicity.

## Conclusion

The ability to psychologically detach from work during the first pandemic lockdown, and improvement in this recovery activity over time, predicted better mental wellbeing and quality of life in vocationally active adults after one year of a global crisis, irrespective of work status. Interventions to encourage workers to psychologically detach from work may help to support employee wellbeing at all times, not only in the extreme circumstances of pandemics and economic uncertainty.

## Introduction

The onset of the COVID-19 pandemic, with rapidly rising infection rates and large-scale lockdown policies introduced to contain the spread of the virus, had a profound effect on the labour market and work activities worldwide [1]. The pandemic evoked a global unemployment crisis [2], and in the United Kingdom (UK), data from the Office for National Statistics shows a sharp decline in working hours from February 2020 (1,052 million hours) to a record low in April 2020 (841 million hours), increasing to 1,000 million hours in 2021 [3]. Inequalities in the labour market have widened, with lockdowns and social distancing impacting in particular on the ability of younger, lower-earning, and less educated people to work [4]. For those remaining in work, there were dramatic changes in the nature of work and work environments, with a transition to home working for many (from 5% in 2019 to 37% in April 2020 sustained to 2021 [5]), travel impacts, and digitalisation [6–8]. The digital and green transitions (referred to as 'twin transition') have been changing the workplace at a rapid pace, leading to new forms of work (e.g., hybrid work, gig economy jobs) or changes in the forms of management and work organisation (e.g., through algorithmic decision-making and digital worker performance monitoring) for workers across the spectrum. The acceleration of these changes coupled with direct impacts of the pandemic on the labour market has affected not only on the economies of the global population, but also the employment reality of a substantial proportion of the vocationally active workforce.

In the early stages of the COVID-19 pandemic, labour market policy initiatives were established across the UK to prevent job losses. In England, for example, the Coronavirus Job Retention Scheme [9] was applied from 1 March 2020 to 30 September 2021, providing grants to employers to ensure that they could retain and continue to pay their workforce through furlough (stopping work completely) or flexible furlough (reduced working hours) agreements with employees, temporarily securing earnings of up to 80% of regular pay. The COVID-19 pandemic, the policy measures to control its spread (e.g., closure of schools, non-essential businesses, many public places and transport systems, lockdowns, physical distancing, social

isolation) and labour market policy initiatives coincided with a marked deterioration in population mental wellbeing [10,11]. Observational population-based studies showed a deterioration of mental health in the UK with the onset and progression of the COVID-19 pandemic and associated lockdowns through 2020 [12–16]. A cohort study of 49 993 participants in 11 longitudinal studies found that the substantial deterioration in mental health observed in the UK during the first lockdown did not reverse when lockdown lifted, but rather worsened across the pandemic period through to March 2021 [17]. Deteriorating or consistently poor mental health has been observed disproportionately in specific groups, such as younger adults and women [17–19], those infected with COVID-19, those with pre-existing health conditions, or with financial difficulties [10,20–22].

During COVID-19 lockdowns and simultaneous economic downturn, induced economic hardship (i.e., substantive loss of income from reduced working hours or job loss) was commonplace [23]. While financial stressors impact negatively on wellbeing [20–22], the relationship between work status for the vocationally active (e.g., employed, self-employed or flexi-furloughed) and wellbeing is variable. For the self-employed, findings from the pandemic contrast with those observed pre-2020. For example, being self-employed has previously been associated with higher job happiness [24–26], even though incomes for the self-employed are lower on average than comparable waged employees [27,28]. Nonetheless, the large and disproportionate reductions in hours and income experienced by the self-employed during COVID-19 directly contributed to a deterioration in their levels of subjective well-being compared to waged workers [29]. These trends were observed across the world where furlough schemes were implemented.

There is little evidence focusing specifically on employees who were flexi-furloughed (reduced working hours) during the COVID-19 pandemic; the flexi-furlough scheme came into effect in July 2020 [30] following the launch of the furlough scheme in March 2020 [31]. While qualitative research suggests that furlough schemes induced uncertainty, anxiety and fear among employees during the pandemic [32], short working hours and furlough job retention schemes showed to be effective protective factors against worsening mental health during COVID-19 (i.e., having some paid work and/or some continued connection to a job was better for mental health than not having any work at all) [33]. Longitudinal data suggests that COVID-19 furlough policy may have mitigated the increase in mental ill-health for some groups of employees [20]. Further, physical health behaviours did not seem to be adversely affected in furloughed workers [34], albeit it is unclear whether workers with reduced hours (flexi-furloughed) were included in this study. Job and financial insecurity [35], changes in job demands [36], increased remote working [37], and expectations regarding accessibility (e.g., from greater use of communication technologies [38]) have all contributed to a blurring of boundaries between work and home lives.

The effort recovery model [39] highlights the importance of resource replenishment in relation to employee mental health and well-being. This model postulates that employees mobilise psychological resources (such as, effort and energy) to engage in job activities and to cope with employment demands and challenges. The process of mobilising the psychological resource is leads to both task performance, but also the risk of depletion over time. Recovery experiences are the mechanism to which employees replenish these depleted or depleting resource is at work [40]. The stressor-detachment model [41] identifies for key recovery activities: psychological detachment (abstaining from thinking about work during non work time); relaxation (maintaining a low activation level); mastery (embracing a positive challenge through learning something new); and control (perceptions of autonomy during non work time) [40]. There is a growing evidence base across all four recovery activities (e.g., [42]) in buffering the potential impact of excessive job stressors in relation to employee mental health (e.g., [42]).

Drawing on the stressor-detachment theory [41], we theorise that detaching psychologically from work may be beneficial in buffering work-related stressors arising from the COVID-19 pandemic and reducing negative consequences for mental health in vocationally active adults. Psychological detachment from work (i.e., mental disengagement from work-related activities, thoughts, problems, and opportunities during off-hours) is known to be beneficial for wellbeing and life satisfaction [43–45] and is understood as an important recovery activity from work [46]. The relationship between psychological detachment, wellbeing and life satisfaction is well established [46]. A meta-analysis [42] observed that higher levels of psychological detachment was associated with a variety of improved personal outcomes, including: mood, energy, compensatory effort, sleep quality, fewer health complaints, wellbeing, and lower stress levels. However, this meta-analysis included studies up until July of 2021, with only a small number of studies included (n = 2) that were longitudinal and spanned the COVID-19 pandemic. The catastrophic impact of the global COVID-19 pandemic on the economy, population wellbeing, and people's work (e.g., job/financial insecurity, income losses, work patterns and location of work) generates a unique circumstance in which the relationships between these variables, in this unprecedented context, and over time, is less clear.

Research on psychological detachment from work during the COVID-19 pandemic is limited, and often focused only on the early months of the pandemic (e.g., April-July 2020), specific occupational groups (e.g., healthcare), or specific aspects of the job (e.g., workload, manager behaviour). For example, psychological detachment from work was found to buffer the negative effects of certain job characteristics in the first three months of the pandemic, such as heavy workload and close monitoring [36]. Other studies, focused on healthcare workers, have shown that psychological detachment from work is positively associated with mental health [47], and is a mediator of within-shift work recovery (in the context of manageable workloads) [48], workload, traumatic stress and work presenteeism [49].

However, researchers have identified a need for longitudinal studies to assess the long-term impact of COVID-19-related changes in work and economic downturn on mental health issues [47]. To our knowledge, the link between psychological detachment from work and subsequent mental health and wellbeing outcomes, after a sustained period of global crisis (with multiple enforced lockdowns), has not yet been determined. Further, the differences in these outcomes between vocationally active adults with varying work circumstances (i.e., employed, self-employed or flexi-furloughed) is not yet known.

Therefore, the aim of this study is to longitudinally explore the relationship between psychological detachment from work, and changes in this recovery activity over time, with subsequent health and wellbeing of vocationally active adults in the UK, within the context of a global pandemic and rapidly changing policy initiatives. The objectives are threefold:

1. to explore self-reported general health, wellbeing (depression, anxiety, life satisfaction), and ability to psychologically detach from work among vocationally active adults, and the change in these outcomes across two waves of data collection (broadly corresponding with UK national lockdowns in 2020 and 2021);

2. to explore whether there were any significant differences in reported health, wellbeing and psychological detachment by work status (employed, self-employed, or flexi-furloughed);

3. to test whether participants' self-reported psychological detachment early in the pandemic (first national lockdown) and the change in this recovery activity over time, predict health and well-being a year later (third national lockdown) in vocationally active adults.

## Methods

### Study design

The Wellbeing of the Workforce (WoW) study was a prospective longitudinal cohort study conducted during the COVID-19 pandemic, with two waves of data collection corresponding with the first and third national lockdowns in the UK. Reporting adheres to the STROBE checklist [50] (S1 Table). The study protocol is registered at: https://doi.org/10.17605/OSF.IO/WA9B3. This study was performed in accordance with the Declaration of Helsinki and all relevant guidelines and regulations. Ethical approval was obtained from the University of Nottingham Faculty of Medicine and Health Sciences Research Ethics Committee in April 2020 (Ref: 03–0420).

### Participants and setting

Participants were working-age adults (over 18 years old) who were employed by an organisation or self-employed in the UK at the start of the study and were able to provide informed consent. There was no maximum age; those exceeding state retirement age but considering themselves to be vocationally active were included. Those who were under 18 years of age, unemployed, furloughed and not working, self-employed and not working, or from employment settings outside of the UK were excluded.

### Procedure

Invitations to take part in the study were shared via diverse distribution channels to minimise response bias, including business-facing professional networks, trade unions, and social media (Twitter, Facebook, and a local TV channel promotional piece) between April-June 2020 (Time 1 [T1]). The invitations contained a link to an online survey (S1 and S2 Texts), hosted on JISC Online Surveys platform. For context, the first survey opened just weeks after COVID-19 was declared a pandemic in the UK and remained open for six weeks during the first national lockdown across the UK [51]. During this first lockdown, the movement and activities of individuals were restricted, such that only organisations supplying basic needs and services were able to function normally.

On clicking the link to the study promotional materials, potential participants accessed an online participant information sheet explaining the purpose of the study, procedures, data storage and how long the survey took to complete, at the end of which they clicked to provide their online consent. Participants were informed that by voluntarily completing and submitting the online survey they were providing their written informed consent to take part. Participants who self-identified as meeting the eligibility criteria, and provided informed online consent, were then able to access the online survey questions. After completion of the first survey, participants were asked to provide their email address if they would like to participate in the follow-up. Participants taking part in the follow-up were given a unique identification code to maintain their anonymity and email addresses were removed from their survey responses. The follow-up survey was administered in March-April 2021 (Time 2 [T2]) and remained open for 5 weeks, during the height of the third national lockdown across the UK [51]. Those who had completed the survey at Time 1 and provided a valid email address, were sent the link to the follow-up survey at Time 2. It should be noted that the flexi-furlough scheme came into effect in July 2020 [52], so we are able to present data for two groups at T1 (i.e., employed, self-employed) and three groups at T2 (i.e., employed, self-employed, flexi-furloughed).

Since reminders can increase response rates in online surveys [53], three reminder emails were sent to participants who had not completed the T2 survey, each one week apart. Data were analysed anonymously. As an incentive to complete the follow-up surveys, those participants who completed all surveys were given the option to enter a prize draw to win a £250 high street shopping voucher. Participants who completed the online surveys were able to provide consent to be contacted with an invitation to take part in a concurrent qualitative interview study; the qualitative findings are reported elsewhere.

## Questionnaire measures

The surveys were compiled by the study team, who had expertise in organisational and health psychology. Items and measures were selected based on the literature and theory [41] and the surveys were pilot tested appropriateness of content, usability, and technical functionality. The surveys took approximately 15 minutes to complete at T1, and seven minutes at T2. The full range of measures collected through this study are provided in S2 Table. In the context of this study, we have only used a relevant selection of these study variables. At T1 (S1 Text), there were 134 questions divided into 8 sections over 10 pages. At T2 (S2 Text), there were 67 questions divided into 8 sections over 10 pages. Items were not randomised or alternated. A routing system within the survey based on work status allowed participants to view only the items that were relevant to them in their circumstances. Participants could review and change their answers through a 'Back' button. Demographic data included age, gender, ethnicity, relationship status, partner work status, living arrangements, working as a key worker, and whether participants had caregiving responsibilities (yes/no). Data were collected on participants' work status (working, furloughed, made redundant due to COVID-19, self-employed and working, self-employed and not working, or other). This study uses data from three of these work status groups: employed and working (T1 and T2), self-employed and working (T1 and T2), flexi-furloughed (T2 only). For this study, the redundant and 'other' groups were excluded due to the focus on vocationally active adults, the small number of cases, and heterogeneity of participants respectively.

**Psychological wellbeing.** The World Health Organisation–Five Wellbeing Index [54] was used to quantify participants' psychological wellbeing. This scale includes five positive worded items using a six-point Likert scale: (0) "at no time" to (5) "all of the time". This scale asks participants to indicate how they have been feeling in the last two weeks. An example item includes "I have felt calm and relaxed". The items are summed together to create a composite score. The raw score ranges from zero (absence of wellbeing) to 25 (maximum wellbeing), but in line with the measure's recommendation, this score is multiplied by four to provide a range of 0 to 100. A lower score is indicative of poor wellbeing. A score of 50 or below is commonly used as a 'screening diagnosis' for depression [55], while a more restrictive score of 28 or below is indicative of 'major depression' [56] This scale demonstrates good psychometric properties in community-based samples [55]. The internal consistency in our sample was good: $\alpha = .87$.

**Anxiety.** The GAD-7 scale [57] was used to quantify participants' general anxiety. The scale includes seven items and asks participants, using a four-point Likert scale, to indicate their experience of anxiety symptoms in the last two weeks. Scaling ranged from: (0) 'not at all' to (3) 'nearly every day'. An example item includes: "Feeling nervous, anxious, or on edge". A higher score is indicative of increased general anxiety. Items are summed together to create a composite score, with a range of zero to 21. Scores of 10 and above are interpreted to indicate moderate to severe levels of anxiety. The scale demonstrates satisfactory reliability and validity within community samples [56,57]. The reliability within our sample was good: $\alpha = .88$.

**Life satisfaction.** A single item was used to measure life satisfaction [58]. The item asks participants "*In general, how satisfied are you with life*?" using a five-point Likert scale: (0) 'very dissatisfied' to (5) 'very satisfied'. Higher scores are indicative of better life satisfaction.

**General health.** A single-item measure of general health was used. This item asks participants to indicate on a scale from (1) 'very bad' to (5) 'very good' the nature of their current health. The item reads: "*How is your health in general*? *Would you say it is...*". A lower score is interpreted as poorer self-rated general health.

**Psychological detachment.** We used the psychological detachment subscale from the Recovery Experience Questionnaire [59]. This subscale includes seven items measured on a five-point Likert scale from (0) 'strongly agree' to (4) 'strongly disagree'. An example item includes "I don't think about work at all". The items were summed together to create a composite score, with a range of 0 to 28. We reverse-coded the items so that lower scores represent lower psychological detachment from work to support interpretation. Previous research has observed satisfactory psychometric properties for this sub-scale internationally [60]. We observed strong internal consistency of this scale within our sample: α = .87.

## Data analysis

Guided by the Checklist for Reporting Results of Internet E-Surveys (CHERRIES) [61] the participation rate was the ratio of those who completed the T2 survey divided by those who completed the T1 survey. Completion rate was the ratio of the number of people who finished each survey divided by those who completed the first page of the survey. No surveys were excluded from analysis; n varied according to number of completers per item. Analysis was undertaken using IBM SPSS Statistics (Version 27). Data cleaning processes were conducted to ensure data were missing completely at random. Of those that responded, less than 5% of data were observed to be missing at the item level. A replace-by-mean strategy was used to address missing data. Outliers were defined as standardised residuals exceeding ± 3.0 [62]. Where identified these cases were removed from relevant analyses. Descriptive statistics were used to explore the prevalence of mental health indicators in our sample at both measurement points. Statistical assumptions underpinning our employed statistical techniques (paired-samples t-test, chi-square and multiple linear regression) were conducted before data analysis. Tests for normality were undertaken at the univariant level using Kolmogorov-Smirnov Tests and Shapiro Wilks test, with skewness and kurtosis statistics presented in a supplementary file (S3 Table). A resampling approach (bootstrapping with 1000 iterations) was used to address potential deviations in normality within study variables. Bias corrected and accelerated (BCa) was specified for 95% confidence intervals [62].

## Results

There were 337 respondents at Time 1 (T1), of which 169 completed the Time 2 (T2) survey; participation rate was therefore 50.15%. Completion rate was 100% at both T1 and T2. The average age of our participants at T1 was 43.91 (11.25, range: 20 to 70, n = 337) and at T2 was 45.47 (10.94; range: 21 to 70; n = 169). A post-hoc power analysis was conducted using G Power (Faul et la., 2007). At time 1, the study had 45% power to detect a small effect size (w = 0.1), 99% power to detect a medium effect size (w = 0.3) and 100% power to detect a large effect size (w = 0.5). At Time 2, the study had 25% power to detect a small effect size, 97% power to detect a medium effect size and 99% power to detect a large effect size. Only two participants exceeded state retirement age, but they reported being vocationally active so remained in the sample. At T1, most respondents identified as female (83%) and White (95%) with 59% located in the Midlands of England. The largest proportion of respondents were

married / civil partnerships (55%), and just over a third had caring responsibilities (39%). Many respondents had a full-time permanent employment contract (59%). At T2, most of these demographics remained broadly consistent, with only slight increases in the proportion of females and respondents identifying as White. See Table 1 for an overview.

## Self-reported health, wellbeing, and psychological detachment from work: Prevalence, difference by work status groups, and changes over time

Descriptive statistics are provided by each health and wellbeing indicator for the total sample and aggregated by each work status group (employed, self-employed) at T1 (N = 337) and T2 (N = 169). We sought to identify the proportion of those 'at risk' at each time point: T1 (1st national lockdown) and T2 (third national lockdown). Those 'at risk' were classified using established scale cut-offs among the validated mental health indicators: major depression (WHO-5 [63]; score ≤28), low mood (WHO-5 [63]; screening diagnosis score ≤50) and anxiety (GAD-7 [64]; score ≥10). Among our single items, those participants categorised as 'at risk' were those that indicated their general health was either '*very bad*' or '*bad*', and those that reported their quality of life was either '*very dissatisfied*' or '*dissatisfied*'. In relation to participants' psychological detachment from work, we created a binary variable by stratifying our group one standard deviation below the mean at T1, to indicate low psychological detachment. This analytical categorisation approach has been used previously within the psychological detachment literature (e.g., [65]).

A series of chi-square tests were used to investigate group differences by work status grouping using these binary variables as dependent variables at T1 and T2. At T2, the flexi-furloughed working group was excluded due to small number of participants (n = 2). Table 2 provides a summary of:

1. the observed means across the study variables for the total sample (T1 and T2);

2. the proportion of those 'at risk' for both the total sample and aggregated by work status groups (T1 and T2); and

3. the results of the series of chi-square tests investigating differences in work status groupings.

**Psychological wellbeing.** During the first national lockdown, almost half (48.1%; 162 cases; N = 337) of the sample, at T1, reported poor psychological wellbeing, meeting the criteria for depression (scoring ≤50). An estimated fifth of our sample met the more restrictive threshold for major depression (20.2%, N = 68, scoring ≤28). At T2, the proportion reporting depression increases to 57.4% (97 cases; N = 169) by T2; with 29% (N = 49) of these meeting the more restrictive threshold for major depression. This was not different by work status groups at T1 or T2. On average, participants' psychological well-being worsened over time: $t$ (164) = 3.55 $p$ = .002; $\chi^2_{diff}$ = 4.66 (2.15 to 7.02; n = 165). The magnitude of this change was small (d = .28;.12 to .43).

**Anxiety symptoms.** Over a quarter (26.1%; N = 337) of the total sample at T1 and almost one third (30.2%; N = 169) at T2 had moderate-to-severe levels of anxiety, scoring ≥10. At both time points, we did not observe a significant difference across the work status groups. We did not observe a statistically significant change in participants' self-reported anxiety from T1 to T2: $t$ (165) = -1.02, $p$ = .15, $\chi^2_{diff}$ = -.38 (-1.15 to .35), d = -.10 (-.25 to .04); n = 179.

**Life satisfaction.** Fifteen per cent (N = 337) of our total sample at T1 and 25.4% at T2 (N = 169) reported being either 'very dissatisfied 'or 'dissatisfied' with their quality of life. We

**Table 1. Sample characteristics at each time point.**

| | Time 1 (N = 337) | | Time 2 (N = 169) | |
|---|---|---|---|---|
| | Frequency[a] | Percentage (%) | Frequency[a] | Percentage (%) |
| **Gender** | | | | |
| Female | 279 | 83.0 | 145 | 86.3 |
| Male | 54 | 16.1 | 22 | 13.1 |
| Non-binary | 3 | .9 | 1 | .6 |
| **Ethnicity** | | | | |
| White | 316 | 95.2 | 160 | 96.4 |
| Asian | 8 | 2.4 | 2 | 1.2 |
| Black/ African/Caribbean | 1 | .3 | - | - |
| Mixed/ Multiple ethnic groups | 7 | 2.1 | 2 | 1.2 |
| Other ethnic groups | - | - | - | - |
| **Location** | | | | |
| England–Midlands | 198 | 59.3 | 104 | 61.9 |
| England–South | 40 | 12.0 | 20 | 11.9 |
| England–North | 41 | 12.3 | 17 | 10.1 |
| England–London | 22 | 6.6 | 9 | 5.4 |
| England–East | 17 | 5.1 | 9 | 5.3 |
| Scotland | 8 | 2.4 | 4 | 2.4 |
| Wales | 4 | 1.5 | 2 | 1.2 |
| Northern Ireland | 3 | .9 | 3 | 1.8 |
| **Relationship Status** | | | | |
| Married/ In a civil partnership | 186 | 55.4 | 93 | 55.4 |
| Single | 103 | 30.7 | 47 | 28.0 |
| Divorced/Separated/Widowed | 32 | 9.5 | 18 | 10.7 |
| Prefer not to say | 15 | 4.5 | 6 | 6.0 |
| **Caring Responsibilities** | | | | |
| No | 203 | 60.4 | 106 | 62.7 |
| Yes | 133 | 39.4 | 63 | 37.3 |
| **Work status** | | | | |
| Employed and working | 306 | 90.8 | 149 | 88.2 |
| Self-employed and working | 31 | 9.2 | 18 | 10.7 |
| Employed and flexi-furloughed[b] | | | 2 | 1.2 |
| **Nature of Employment Contract** | | | | |
| Full-time permanent | 198 | 58.8 | 93 | 66.0 |
| Part-time permanent | 68 | 22.1 | 27 | 19.1 |
| Full-time fixed term | 21 | 6.8 | 10 | 7.1 |
| Part-time fixed term | 12 | 3.9 | 8 | 5.7 |
| Other[c] | 8 | 2.4 | 3 | 2.1 |
| **Key/Essential worker** | | | | |
| Yes | 107 | 34.9 | 35 | 25.0 |
| No | 198 | 64.5 | 104 | 74.3 |
| Prefer not to say | 2 | .7 | 1 | .7 |

[a]Completion was non-mandatory, therefore, total frequency varies by item.

[b]Data available from T2 only. During flexi-furlough, employers paid employees at their usual rate of pay for the hours worked and any remaining days or hours (i.e., furloughed days/hours), were paid under the furlough scheme, subject to relevant cap.

[c]e.g., zero hours, seasonal hours only.

**Table 2. Proportion of those 'at risk' of poor health (%, valid cases/ sub-group sample) and wellbeing at T1 (N = 337) and T2 (N = 178) and testing for differences by work status group by time period.**

| | | Total Sample ($N_{T1}$ = 337; $N_{T2}$ = 169) | | Employed and working | Self employed | Flexi-Furloughed*** | Chi-square Test* |
|---|---|---|---|---|---|---|---|
| | | Mean (95%CI; SD); range | % (cases/ total sample) | | | | |
| 1. Major Depression (WHO-5 $\leq$28) | T1 | 49.68 (20.03); 4 to 96* | 20.2 (68/337) | 19.9 (61/306) | 22.0 (7/31) | N/A | $\chi^2$ (1) = .122, $p$ = .72, Cramer's V = .02 |
| | T2 | 44.62 (21.16); 0 to 100 | 29.0 (29/169) | 28.9 (43/149) | 27.8 (5/18) | 50% (1/2) | $\chi^2$ (1) = .40, p = .84, Cramer's V = .02 |
| 2. Screening level depression (WHO-5 $\leq$50) | T1 | Same as above. | 48.1 (162/337) | 49.0 (150/306) | 38.7 (12/31) | N/A | $\chi^2$ (1) = 1.20, p = .27, Cramer's V = .06 |
| | T2 | Same as above | 57.4 (97/169) | 57.0 (85/149) | 55.6 (10/18) | 100 (2/2) | $\chi^2$ (1) = .00, p = .95, Cramer's V = .01 |
| 2. Moderate to severe anxiety (GAD-7 $\geq$10) | T1 | 6.94 (5.45); 0 to 21 | 26.1 (88/337) | 26.1 (80/306) | 25.8 (8/31) | N/A | $\chi^2$ (1) = .002, p = .97 Cramer's V = .00 |
| | T2 | 7.11 (5.42); 0 to 21 | 30.2 (51/169) | 30.9 (46/149) | 27.8 (5/13) | 0 (0/2) | $\chi^2$ (1) = .00, p = .98, Cramer's V = .00 |
| 3. Poor life satisfaction | T1 | 2.7 (.98); 0 to 4 | 15.4 (52/337) | 16.0 (49/306) | 9.7 (3/31) | N/A | $\chi^2$ (1) = .87, p = .35, Cramer's V = .05 |
| | T2 | 2.24 (1.00); 0 to 4 | 25.4 (23/169) | 25.5 (38/149) | 22.2 (4/18) | 50 (1/2) | $\chi^2$ (1) = .00, p = 97, Cramer's V = .00 |
| 4. Poor general health | T1 | 4.89 (.85); 2 to 6 | 4.7 (16/337) | 3.6 (11/306) | 16.1 (5/31) | N/A | $\chi^2$ (1) = 9.78, p = .002, Cramer's V = .17 |
| | T2 | 4.64 (.82); 2 to 6 | 0 (0/169) | 0.0 (0/149) | 0 (0/18) | 0 (0/2) | ** |
| 5. Low psychological detachment from work | T1 | 4.66 (3.44); 0 to 16 | 21.4 (72/337) | 20.9 (64/306) | 25.8 (8/31) | N/A | $\chi^2$ (1) = .40, p = .54, Cramer's V = .03 |
| | T2 | 7.31 (3.78) 0 to 16 | 16.0 (27/169) | 7.5 (11/149) | 5.01 (1/18) | 0 (0/2) | $\chi^2$ (1) = .31, p = .85, Cramer's V = .04 |

*Composite measure used mean

** Chi-square cannot be calculated with an empty cell

*** due to small sample this work status was excluded from chi-square testing.

did not observe significant differences by work status at T1 and T2. We observed a small (d = .25;.01, .41), but statistically significant decrease in participants' life satisfaction from the first to third national lockdown: t (164) = 3.24, p = .002; $\chi^2_{diff}$ = .24 (.01 to .36); n = 165.

**General health.** A small proportion of the sample reported poor general health (4.7%; N = 337) at T1 and (0%; N = 169) at T2. At T1, we observed a statistically significant difference by work status group in relation to the observed proportionality of those with poor self-reported general health ($\chi^2$ = 9.78, $p$ = .002). We can observe that those participants in the 'self-employed and working' work status group comprised, on average, a larger proportion of those indicating poor levels of self-reported general health (16.1%) as compared to the other groups. We observed a small (d = .29, 14, .45), but statistically significant, reduction in participants' self-reported general health from the first to third national lockdown: t (164) = 3.74, $p$ < .001; $\chi^2_{diff}$ = .22 (.09 to .35); n = 165.

**Psychological detachment from work.** Participants one standard deviation below the mean was used to provide a threshold to quantify low psychological detachment from work at T1 (low psychological detachment $\leq$1.22). One-fifth of the sample (21.4%; N = 337), at T1, reported low psychological detachment from work. At T2, we observed 7.1% (N = 169) to report low psychological detachment from work. We did not observe any differences by work status at T1 or T2 by time period.

**Table 3. Pearson product-moment correlation matrix.**

| | 1 | 2 | 3 | 4 | 5 | 6 | 7 | 8 | 9 |
|---|---|---|---|---|---|---|---|---|---|
| 1. Age | 1 | | | | | | | | |
| 2. Gender[+] | -.037 | 1 | | | | | | | |
| 3. Ethnicity | .138 | .091 | 1 | | | | | | |
| 4. Psychological Detachment: T1 | .031 | .226** | -.061 | 1 | | | | | |
| 5. Change in Psychological Detachment from T1 to T2 | -.07 | -.025 | -.016 | .438** | 1 | | | | |
| 6. Psychological Wellbeing: T2 | .145 | -.147 | -.047 | .182* | -.179* | 1 | | | |
| 7. Anxiety: T2 | -.217* | .032 | .005 | -.324*** | .085 | -.575*** | 1 | | |
| 8. Life Satisfaction: T2 | .128 | -.027 | -.103 | .217* | -.123 | .704*** | -.559*** | 1 | |
| 9. General Health: T2 | .064 | -.14 | -.08 | .086 | -.066 | .363*** | -.304*** | .446*** | 1 |

*$p < .05$

**$p < .001$

*** $p < .001$, n = 136; [+] Kendall's Tau correlations calculated for correlation using gender (dummy coded, with women as reference group).

To explore change over time across these variables we conducted a series of paired sample t-text using our repeat measure sample by work status group (employed and working and self employed). See Table 3.

We tested whether participants' ability to psychologically detach from work changed over time. We found a statistically significant mean change from T1 (X = 4.62, SD = 3.36) to T2 (X = 7.07, SD = 3.91), with participants' ability to psychologically detach from work improving over this period: $t$ (129) = -7.09, $p < .001$, $\chi^2_{diff}$ = -2.45, -3.11 to -1.90, n = 130). The magnitude of this change over time was moderate in nature (d = -.62, -.8, -.43). However, when aggregated by work status, this relationship was only significant in the 'employed and still working' group (n = 119) and was not significant in the 'self-employed' group (n = 11). However, it is important to note that due to the small sample size of self-employed participants in our repeat measure sample it is likely this test is underpowered and should be interpreted with caution.

We explored whether there was a change in study variables by work status group (employed and working, self-employed) available within our repeat measure sample (Table 3).

## Psychological detachment from work as a predictor of health and wellbeing

A series of Pearson's product moment correlation coefficients were conducted to examine the relationship between the independent (psychological detachment at T1; change in psychological detachment from T1 to T2) and dependent (psychological wellbeing, anxiety, life satisfaction, and self-reported general health all at T2) variables. The independent variable seeking to quantify the change of psychological detachment from work over time was created by subtracting the T2 from the T1 score to create a new distribution (Table 3).

We conducted a series of multiple linear regressions to test the predictive associations between participants' self-reported ability to mentally switch off from work at T1, and its change over time, and their health and wellbeing later in the pandemic (T2) after controlling for age, gender, and ethnicity. We built the regression models by entering the covariates in block 1 (forced entry), then the participants' self-reported psychological detachment at T1 (block 2; forced entry), followed by the change in their psychological detachment over time (block 3; forced entry). We specified the dependent variable as the T2 measure of participants' major depression, anxiety, life satisfaction, and self-reported general health. Tables 4 and 5 provide an overview of the observed statistics within these regression analyses.

**Table 4. Linear models of predictors of psychological wellbeing (major depression) and anxiety symptoms at T2.** 95% Bias corrected and accelerated confidence intervals reported in parentheses. Confidence intervals and standard errors based on 1000 bootstrapped samples.

| | | Psychological Wellbeing (T2; n = 127) | | | | Anxiety (T2; n = 127) | | | |
|---|---|---|---|---|---|---|---|---|---|
| | | B (95% CI) | SE B | β | p | B (95%) | SE B | β | p |
| Model 1 | Constant | 42.92 (8.46, 86.69) | | | 0.01 | 10.19 | 5.09 | | .003 |
| | Age | 0.30 (.00, .60) | .15 | .156 | 0.05 | -0.11 (-.21, -.01) | .050[b] | -0.22 | .03 |
| | Gender[1] | -5.77 (15.40, 2.92) | 4.49 | -.1 | 0.19 | 0.84 (-1.86, 4.40) | 1.552[b] | 0.06 | .57 |
| | Ethnicity | -2.94 (-13.77, 5.21) | 5.2 | -.06 | 0.41 | 0.53 (-.95, 4.63) | 1.221[b] | 0.04 | .40 |
| | ΔR² = .04, p = .167 | | | | | ΔR² = .05, p = .08 | | | |
| Model 2 | Constant | 34.95 (-10.23, 63.89) | 18.44 | | 0.006 | 13.393 | 5.48 | | .002 |
| | Age | 0.27 (.00, .55) | .14 | .14 | 0.07 | -.10 (-.20, -.01) | 0.045 | -0.19 | .03 |
| | Gender | -8.34 (-17.96, .76) | 4.56 | -.15 | 0.07 | 1.87 (-.86, 5.58) | 1.62 | 0.13 | .22 |
| | Ethnicity | -1.92 (-9.15, 9.45) | 4.52 | -.04 | 0.57 | .12 (-3.63, 2.38) | 1.32 | 0.01 | .86 |
| | Psychological detachment (T1) | 1.31 (.22, 2.32) | .53 | .21 | 0.01 | -.524 (-.80, -.25) | 0.14 | -0.31 | .001 |
| | ΔR² = .04, p = .02 | | | | | ΔR² = .09, p < .001 | | | |
| Model 3 | Constant | 26.85 (-19.14, 53.90) | 18.69 | | 0.04 | 15.03 | 5.19 | | .001 |
| | Age | 0.19 (-.01, .46) | .14 | .10 | 0.2 | -0.08 (-.17, .01) | 0.046 | -.16 | .07 |
| | Gender | -9.31 (-18.19, -.035) | 4.47 | -.17 | 0.04 | 2.06 (-.61, 5.62) | 1.56 | .14 | .16 |
| | Ethnicity | -1.28 (-8.28, 9.82) | 4.54 | -.03 | 0.73 | -0.01 (-3.70, 1.79) | 1.23 | .00 | .98 |
| | Psychological detachment (T1) | 2.30 (1.18, 3.28) | .53 | .36 | < .001 | -0.72 (-1.01, -.42) | 0.148 | -0.43 | .001 |
| | Change in psychological detachment over time | 1.95 (.95, 2.89) | .00 | .36 | < .001 | -0.39 (-.64, -.15) | 0.124 | -0.27 | .002 |
| ΔR² = .11, p = ≤.001 | | | | | | ΔR² = .06, p = .003 | | | |

[1]Women, reference group.

**Psychological wellbeing.** A total of 15.4% (adjusted $R^2$) of the variance of psychological wellbeing as reported during the third national lockdown (T2) was accounted for by the covariates: participants reporting an ability to detach from work at T1, and the change in this recovery activity over time. Among our covariates, gender was observed to be statistically significant (β = -.17, $p$ = .04), with men, in comparison to women (the reference group), .17 below women. Among our independent variables, we observed both to be statistically significant predictors of participants' psychological wellbeing at T2. More specifically, we observed that, on average, the more participants engaged in psychological detachment from work at T1 the better their self-reported psychological well-being (i.e., lower risk of depression) at T2 (β = .21, $p$ = .01). In addition, we observed that improvement over time in this recovery activity corresponded to, on average, better psychological well-being (i.e., lower risk of depression) at T2 (β = .36, $p$ < .001).

**Anxiety symptoms.** A total of 17.1% (adjusted $R^2$) of the variance in anxiety symptoms during the third national lockdown was accounted for by our covariates and two independent variables. In our final regression model, none of our covariates were statistically significant. We observed that, on average, lower levels of anxiety at T2 were associated with psychological detachment from work at T1 (β = -.43, $p$ < .001) and the improvement of this recovery activity over time (β = -.27, $p$ < .001).

**Life satisfaction.** An estimated 10.7% (adjusted $R^2$) of the variance in self-reported life satisfaction in the third national lockdown was accounted for by our covariates and two

**Table 5. Linear models of predictors of life satisfaction and general health at T2.** 95% Bias corrected and accelerated confidence intervals reported in parentheses. Confidence intervals and standard errors based on 1000 bootstrapped samples.

| | | Life Satisfaction (T2; n = 127) | | | | General Health (T2; n = 127) | | | |
|---|---|---|---|---|---|---|---|---|---|
| | | B (95% CI) | SE B | β | p | B (95%) | SE B | β | p |
| Model 1 | Constant | 2.83 (.41, 5.45) | 1.01 | | .001 | 4.99 (3.80, 7.79) | .73 | | .001 |
| | Age | .02 (.00, .03) | .01 | .19 | .01 | .01 (-.01, .02) | .01 | .11 | .18 |
| | Gender[1] | .03 (-.50, .46) | .25 | .01 | .89 | -.32 (-.65, .02) | .17 | -.14 | .05 |
| | Ethncity | -.33 (-.87, .16) | .25 | -.13 | .04 | -.18 (-.76, .07) | .19 | -.08 | .01 |
| | $\Delta R^2$ = .05, p = .116 | | | | | $\Delta R^2$ = .04 p = .16 | | | |
| Model 2 | Constant | 2.46 (-.27, 3.45) | .94 | | .001 | 4.88 (3.2, 7.46) | .75 | | .001 |
| | Age | .02 (.00, .03) | .01 | .18 | .03 | .01 (-.01, .02) | .01 | .11 | .21 |
| | Gender | -.09 (-.64, .34) | .26 | -.03 | .72 | -.35 (-.71, .03) | .19 | -.16 | .05 |
| | Ethnicity | -.28 (-.67, .52) | .23 | -.11 | .05 | -.16 (-.70, .12) | .18 | -.08 | .10 |
| | Psychological Detachment (T1) | .06 (.02, .11) | .02 | .20 | .01 | .02 (-.04, .07) | .03 | .07 | .50 |
| | $\Delta R^2$ = .04, p = .03 | | | | | $\Delta R^2$ = .01, p = .43 | | | |
| Model 3 | Constant | 2.17 (-.70, 3.23) | .98 | | .003 | 4.76 (3.14, 7.25) | .72 | | .001 |
| | Age | .01 (-.00, .03) | .01 | .14 | .08 | .01 (-.01, .02) | .01 | .09 | .28 |
| | Gender | -.12 (-.60, .30) | .24 | -.05 | .63 | -.37 (-.72, .02) | .18 | -.16 | .05 |
| | Ethncity | -.26 (-.57, .52) | .24 | -.11 | .06 | -.15 (-.61, .14) | .18 | -.07 | .10 |
| | Psychological Detachment (T1) | .10 (.04, .15) | .03* | .32 | .003 | .03 (-.03, .09) | .03 | .13 | .26 |
| | Change in psychological detachment overtime | .07 (.02, 12) | .03* | .27 | .008 | .03 (-.01, .07) | .02 | .13 | .21 |
| $\Delta R^2$ = .06, p = .005 | | | | | $\Delta R^2$ = .01, p = .19 | | | | |

[1] women, reference group.

psychological detachment variables. In our final regression model (model 3), we observe that none of the covariates were significant predictors. Again, we observed that better psychological detachment from work at T1 (β = .32, *p* = .003) and the improvement in this recovery activity over time (β = .27, *p* = .008) were both associated with better life satisfaction as reported in the third national lockdown.

**Self-reported general health.** An estimated 2% (adjusted $R^2$) of the variance in self-reported general health at T2 was accounted for by the covariates and two psychological detachment variables. None of the covariates or independent variables tested within this regression model was found to significantly predict self-reported general health at T2, although gender approached statistical significance.

Predictors of psychological wellbeing and anxiety at T2 are shown in Table 4. Predictors of life satisfaction and health are shown in Table 5.

## Discussion

This is the first study, globally, to longitudinally explore the relationship between psychological detachment from work, and changes in this recovery activity across two waves of data collection, with subsequent health and wellbeing of working-age adults.

Our study was undertaken within the extreme context of a global pandemic and rapidly changing COVID-19 mitigation policy initiatives which impacted negatively on population wellbeing [12–16]. Our data collection waves coincided with the first surge of COVID-19 in the UK and immediate lockdown from late March 2020 (T1) and the third lockdown in March-April 2021 (T2). Irrespective of work status, psychological wellbeing (major depression), general health ratings and life satisfaction deteriorated over time, while levels of anxiety were moderate-to-high at the outset of the pandemic, and this was sustained after a year.

Supporting the stressor-detachment theory [41], higher psychological detachment from work in the early months of the pandemic, and the improvement in this recovery activity over time, while unrelated to general health, predicted better psychological wellbeing (i.e., lower risk for depression), lower anxiety and higher life satisfaction after one year.

Almost half our sample in the first wave, rising to over 57% in the second wave, reported low psychological wellbeing (i.e., depression at 'screening diagnosis' level), with one fifth reporting indicators of major depression in the first wave (based on the more stringent WHO-5 criteria), rising to almost one third in the second wave. The marked decline in psychological wellbeing over time was observed in those with more, or less severe symptoms of depression, and this pattern concurs with findings from general population studies conducted in the UK during the pandemic, albeit with variations in the measurement scales used and data collection time points [10,12–17]. This is likely to be directly associated with the impacts of the pandemic, since analysis of nationally representative data shows that mental health issues identified during the COVID-19 pandemic are unlikely to be associated with seasonality or variation year-to-year [14].

High anxiety was prevalent in our sample—cross-sectional studies conducted in the general population early in the pandemic also identified anxiety levels that exceeded population norms [19]. The proportion of our participants experiencing moderate-to-severe anxiety on the GAD-7 was high (one quarter rising to one third, across waves) and higher than that observed in the UK COVID-19 Mental Health and Wellbeing study (UK COVID-MH) [12] (reporting 17.2% and 16.7% with moderate anxiety at broadly similar time points). This may be partially explained by the higher drop-out of participants with high anxiety in the UK COVID-MH study, and differences in sample demographics; including adults of any age, compared to working-age adults only. As with UK COVID-MH, high anxiety levels in our sample were sustained over time. This could demonstrate that vocationally active adults experienced heightened anxiety throughout this pandemic period (i.e., from first to third lockdown). Alternatively, anxiety levels may have fluctuated during this time (i.e., worse during lockdowns, improving as lockdowns eased), since the COVID-19 Social Study showed that restrictions that infringed more on social freedoms (i.e., lockdowns) were found to have more damaging effects on mental health [10].

Patterns observed in our data (i.e., decline in psychological wellbeing and high anxiety) were irrespective of work status, with no differences between those who were employed or self-employed. Although, pre-pandemic, financial worries were found to be higher among the self-employed due to the volatility and instability associated with self-employment [66,67], analysis of data from the Understanding Society's COVID-19 survey of the UK population conducted in April 2020 [68] found no differences in financial worries between self-employed and employed participants [69]. It is possible, that financial concerns were not a factor in the wellbeing of our sample, although this is unlikely since financial adversity during the pandemic (e.g., reductions in household incomes and increased cost of living) has been consistently associated with mental ill-health during this time [20,22,70]. In fact, the COVID-19 Social Study found negative effects on mental health not only from experience of adversity, but also the worry about potentially experiencing it [10].

We therefore advocate, as proposed by Wolfe and Patel, that the lack of differences in psychological wellbeing and anxiety between different work status groups may simply point to the magnitude of the effects of the pandemic across the population [69], with negative impacts on all vocationally active adults, irrespective of employment status (i.e., employed or self-employed). Notably, we had a high proportion of key workers in our sample (37% at T1, 25% at T2). This may partially explain the prevalence of poor psychological wellbeing and high anxiety in our sample, since the COVID-19 Social Study found that keyworkers (particularly those

in the essential services category: utility, food chain and transport roles) had consistently higher levels of depressive and anxiety symptoms than non-keyworkers across the whole of the study period [71].

We found that life satisfaction dropped over time in our sample of participants, albeit the effect size was small. Although there are myriad reasons why this might be, it is not entirely unexpected, since research supports an 'accumulation hypothesis' in women who are mothers [72]. The accumulation hypothesis suggests that pandemic stressors accumulate, leading to even lower satisfaction over time; our sample was 84.8% female and around one third reported caregiving responsibilities. Our sample was working age; international studies show that life satisfaction declined more in younger adults compared to those of 60 years or older (e.g., Canada [73]). Nationally representative data shows lower levels of population life satisfaction during the COVID-19 pandemic, compared to pre-pandemic levels [2,16]. By April-June 2021, alongside easing of lockdowns and advancement of COVID-19 vaccination in the UK, population data shows that life satisfaction had returned to pre-pandemic levels for many, although there was no improvement identified in certain age groups, that comprise a large proportion of working-age adults (i.e., people under 25, those between 35 and 39) [74].

General health declined in our sample from the first to second wave of data collection which is not unexpected given escalating COVID-19 infection rates, and negative changes in lifestyle behaviours observed during the pandemic and associated lockdowns (e.g., decreased physical activity, increased sedentary behaviour [75–77], poor dietary habits and increased alcohol consumption [78]). It is notable that, in our study, only a minority reported poor general health at each time point. However, the impact of the pandemic has not been equal across society, and groups that were more affected were often the more vulnerable, and had worse health prior to the pandemic [79]. Our survey participants were a relatively young demographic, the majority of whom were working at the time of the study, and therefore it is possible that the survey was not completed by those in society with the poorest physical health. The evidence surrounding those who are self-employed is mixed. Early research showed higher levels of behavioural and physiological risk factors among the self-employed than among salaried workers [80], but others have found that self-employed people are as healthy as wage-earners and more likely to engage in healthy behaviours [81]. We found that self-employed participants in our study were more likely to report poor general health than those who were employed. This perception of poor health may reflect specific pandemic impacts on those who are self-employed (e.g., work volatility, job insecurity, reductions in hours and income) which has shown to impact on wellbeing [29] and may similarly impact on perceived general health. Analysis of work interruption data in the US shows that self-employers were hit harder by the COVID-19 pandemic and recovered more slowly [82]; this is likely to have induced work-related stress which is known to be associated with poorer self-perceived health and increased physical illness symptoms [83].

Our findings build on the literature related to the stressor-detachment model [41]. During the first national lockdown, over one fifth of our sample reported low detachment from work. Although home working had been increasing during the years preceeding the COVID-19 crisis [84], the proportion of people working exclusively from home rose dramatically and suddenly during the first lockdown, increasing eight-fold from 5.7% of workers in January/February 2020 to 43.1% in April 2020 [85]. Studies have shown that, in those who worked from home during the pandemic lockdowns, taking fewer rest breaks is linked to decreased psychological detachment from work [86]. This has practical implications for educating line managers about the importance of job design to allow sufficient rest breaks, and educating employees about the value of job crafting (i.e., individuals taking proactive steps to change characteristics of their job) which could, for example, include monitoring and regulating their rest break

behaviour while remote working, and taking actions to adjust job demands, shape job activities and adapt ways of working to better facilitate adequate rest breaks.

Qualitative research has identified factors associated with low detachment from work for home workers, such as increased technology use, working longer hours, expected availability outside of office hours and checking emails late evening, and poor role modelling of supervisors; a lack of subsequent detachment from work and associated 'cognitive weariness' then impacting negatively on wellbeing [87]. There may be scope to further explore segmenting strategies (i.e., establishing boundaries between work and non-work) as a tool for enabling recovery while working from home [88]). A recent meta-analysis including 30 studies reporting 34 interventions (in papers published up to June 2020) demonstrates that interventions addressing job stressors or altering primary and secondary appraisal can have positive effects on detachment from work, regardless of how detachment is conceptualised [89].

The turbulent context of a pandemic is relevant here, and for many, the transition to working from home was sudden and unanticipated. Although evidence from UK longitudinal populations surveys indicates that home working per se, does not have a lasting detrimental impact on wellbeing [90], it was suggested that psychological detachment is an important early indicator of an employee's ability to successfully manage the interface of work and family domains during the transition to remote working during the lockdown [91]. This may help to explain our finding that psychological detachment from work in the first lockdown, and the improvement in this recovery activity over time predicted better outcomes in terms of psychological wellbeing and quality of life by the third lockdown, one year later. These findings were substantiated across all work groups. We observed an overall improvement in the mean scores for detachment between T1 and T2, at absolute levels and within our repeat sample. Such improvements in psychological detachment from work may reflect the use of positive coping strategies by those who continued to work from home, or gradually returned to hybrid working (e.g., establishing work boundaries and improving work-life balance). Examples of positive strategies come from the COVID-19 Social Study, in which improvements in mental health were seen among people who spent time outdoors and in green space, those who connected socially with friends and family, and those who engaged in leisure pursuits such as exercise, hobbies and creative activities [10].

A study conducted in the US found that perceived work-life balance improved during the pandemic compared to before–the hybrid workplace was seen to offer a level of flexibility that was positively associated with work productivity, satisfaction, and work-life balance [92]. Adoption of positive coping strategies, and the flexibility of hybrid working may have facilitated psychological detachment from work, for some.

## Strengths and limitations

The prospective cohort design allowed for a 'real-time' study of multiple outcomes during multiple exposures to a unique ('rare') and extreme national phenomenon, with COVID-19 lockdowns occurring across the UK, at different stages of the pandemic [51]. We were able to explore workers' experiences in the context of rapid changes in the national and global pandemic situation and were therefore able to explore the dynamic relationship between outcome measures, and the national lockdowns. However, the longitudinal nature of the design, coupled with the global uncertainty during this period, increased vulnerability of the study to a high rate of loss to follow-up. The broad inclusion criteria allowed for data to be gathered from a diverse pool of workers across sectors, organisation size and types, in areas of more, or less affluence, which enhanced generalisability of the findings. However, although responses were gathered from different genders, ethnic groups, and geographical regions, most respondents

were White and female, with the highest proportion based in the Midlands of England. This paper was focused on vocationally active adults who responded to surveys in the WoW study and so we did not include people who were unemployed, or those who were furloughed in our analyses. While we included participants who were flexi-furloughed as they were vocationally active, the numbers of participants in this work group at T2 was too low to allow meaningful comparison with other work status groups. Since our sample did not include furloughed workers, future research might seek to explore the unique experiences of this group. Other studies have shown that, while furloughed workers experienced anxiety during the COVID-19 pandemic [32], furlough job retention schemes appeared to be protective against worsening mental health during the crisis [20,33].

To test the observed associations we utilise a linear regression technique, while analytically robust we acknowledged the potential value of using multi level modelling. Yielding, potentially, increased precision in the estimated standard errors regression coefficients [93]. Our regression model only accounted for a small number of covariates, this was to keep the model parsimonious and to maintain as much statistical power as possible given our smaller sample size. However, it may be advantageous to explore wider number of demographic and employment variables within this analysis.

Additionally, we acknowledge the potential analytical limitations of dichotomising of psychological detachment variables using a standard deviation below the mean, rather than above or below the mean. By utilising this dichotomisation approach–of focusing on 'extreme cases' we have not captured participants reporting average and slightly below levels of psychological detachment and if and how those changed over time and by employment status.

## Study implications

Our study highlights that the experience of working through a pandemic took its toll on the mental health of both employed, and self-employed workers. However, it clearly demonstrates that strategies such as psychological detachment from work can be protective over time, particularly during challenging times. Research undertaken prior to the pandemic demonstrates the key role of workplace policy, culture and norms, job characteristics, the commute to work, and individual factors in detachment from work. For example, while segmenting work and nonwork roles is known to help employees detach and recover from work demands, segmentation norms within a work group are associated with individuals' experiences outside of work [94]. Therefore, workplace culture (and policy) plays a key role in encouraging or discouraging segmentation between work and nonwork life, and ultimately disengagement (or not) from work. There is a growing literature on the function of commutes as work-home transitions that assist with the process of psychological detachment (and recovery from) work [95]. Job-related variables such as high workload are associated with lower psychological detachment [45,96]. Individual factors influence psychological detachment (or the impact of low detachment) in various ways; emotion regulation plays a role, since high emotional rumination (i.e., past-oriented tendency to be preoccupied with emotional upset) has been found to predict difficulties in psychological detachment from work, and relaxation [97]. High performance-based self-esteem can be a barrier to psychological detachment [98], although low detachment appears to be less detrimental in those with high levels of autonomous work motivation [99]. Such studies indicate that psychological detachment can be more, or less challenging to achieve, and the effects of low detachment can be more, or less damaging to wellbeing for different individuals. There may be a need to target interventions to promote psychological detachment in those for whom it may be most beneficial.

There is a need for research which explores the specific actions (i.e., coping strategies) taken by those who are better able to detach from work, and actions taken by employers,

where relevant. Importantly, the extreme circumstances of several years of a pandemic, and the resulting economic and mental health impacts on vocationally active adults, may have led to longer-term changes in the way in which individuals view work, and how they want to work, and how that is aligned with workers' psychological wellbeing. This paves the way for research which investigates the impacts of remote and hybrid working which proliferated during and after the pandemic, and the relationship between these 'new ways of working' and psychological detachment, mental wellbeing, and work productivity. Exploring the link between psychological detachment from work and work productivity is likely to be vital in developing the case for employers to promote psychological detachment from work within policy and practice, particularly since detachment is associated with lower presenteeism [91] and the economic burden of productivity loss due to presenteeism is vast, particularly with relation to mental ill-health [100]. In recent years, a four-day working week has been advocated, with arguments that a shorter working week may improve working lives, protect health and wellbeing, and enhance work productivity [101,102]. Although in agreement that a reduction in working hours may have broad benefits, Spencer warns of the caveats of the four-day week and argues that working hours reduction "must be situated in a broader agenda: one that encourages a transformation in—and of—work" [103].

Our study has implications for national and international policy and employment legislation. Disengagement from work is not a COVID-19 specific issue with debates in France on the right to disconnect receiving international attention in the mid-2010s [104]. This led organisations, such as the UK Chartered Institute of Personnel and Development (CIPD) to explore the idea by producing example policy guidance for employers [105] but no national UK policy has been forthcoming until 2023 despite manifesto commitments on 'flexible work requests by default' being advocated by many political parties. The debate may see a resurgence in the UK following a policy speech by the Labour Party in May 2023 [106] in which it was indicated that the party would install a right to restrict employer contact out of hours into their manifesto by Angela Raynor as shadow secretary of state for the future of work, echoing legislation in France and alongside other examples from Scotland and Belgium where organisation policies are being introduced in some sectors or with certain sized businesses to promote the right to disconnect.

## Conclusion

An inability to psychologically detach (that is, mentally switch off) from work, depression and anxiety were prevalent in vocationally active adults during the COVID-19 pandemic lockdowns, over an extended period of time. Anxiety remained high during lockdowns, but psychological wellbeing and life satisfaction deteriorated over time, yet the ability to detach from work had improved by the third lockdown. This longitudinal cohort study shows that psychological detachment from work in the first pandemic lockdown, and improvements in this recovery activity over time, predicted psychological wellbeing and life satisfaction during national lockdown one year later. This pattern was observed irrespective of work status (employed or self-employed). This finding is likely to reflect, first, the enormity of impact of 'unprecedented' times of crisis on all working adults, and second, the high value of psychological detachment from work as an important predictor of psychological wellbeing and quality of life over time. This study highlights that psychological detachment from work plays a protective role for wellbeing in times of crisis and uncertainty. There is a need for organisations and individuals to explore best approaches to detach from work to support employee wellbeing. Our findings inform ongoing policy debates on the right to disconnect.

## Supporting information

**S1 Text. Time 1 survey.**
(PDF)

**S2 Text. Time 2 survey.**
(PDF)

**S1 Table. STROBE statement.**
(DOCX)

**S2 Table. Study variables and measures used over time series data collection.**
(DOCX)

**S3 Table. Testing normality of variables.**
(DOCX)

## Acknowledgments

The authors thank Kristina Newman for review of the data collection materials used in this study.

## Author Contributions

**Conceptualization:** Holly Blake, Juliet Hassard, Louise Thomson, Maria Karanika-Murray.

**Data curation:** Juliet Hassard, Louise Thomson, Wei Hoong Choo, Teixiera Dulal-Arthur.

**Formal analysis:** Juliet Hassard.

**Funding acquisition:** Holly Blake, Juliet Hassard, Louise Thomson.

**Investigation:** Juliet Hassard, Louise Thomson, Wei Hoong Choo, Teixiera Dulal-Arthur, Lana Delic, Richard Pickford, Lou Rudkin.

**Methodology:** Holly Blake, Juliet Hassard, Maria Karanika-Murray.

**Project administration:** Wei Hoong Choo, Teixiera Dulal-Arthur, Lana Delic.

**Writing – original draft:** Holly Blake, Juliet Hassard.

**Writing – review & editing:** Louise Thomson, Wei Hoong Choo, Teixiera Dulal-Arthur, Maria Karanika-Murray, Lana Delic, Richard Pickford, Lou Rudkin.

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
