## [Decision Letter · Decision Letter 0]

24 Jun 2024

PONE-D-23-17342Psychological detachment from work predicts mental wellbeing of working-age adults: findings from the ‘Wellbeing of the Workforce’ (WoW) prospective longitudinal cohort studyPLOS ONE

Dear Dr. Blake,

Thank you for submitting your manuscript to PLOS ONE. After careful consideration, we feel that it has merit but does not fully meet PLOS ONE’s publication criteria as it currently stands. Therefore, we invite you to submit a revised version of the manuscript that addresses the points raised during the review process.

We look forward to receiving your revised manuscript.

Kind regards,

Mihajlo Jakovljevic, MD, PhD, MAE

Academic Editor

PLOS ONE

Journal Requirements:

Reviewers' comments:

Reviewer's Responses to Questions

**Comments to the Author**

1. Is the manuscript technically sound, and do the data support the conclusions?

Reviewer #1: Yes

Reviewer #2: Partly

2. Has the statistical analysis been performed appropriately and rigorously? 

Reviewer #1: Yes

Reviewer #2: I Don't Know

3. Have the authors made all data underlying the findings in their manuscript fully available?

Reviewer #1: Yes

Reviewer #2: No

4. Is the manuscript presented in an intelligible fashion and written in standard English?

Reviewer #1: Yes

Reviewer #2: Yes

5. Review Comments to the Author

Reviewer #1: Overall this has been a comprehensive paper, covering key issues around the lockdown time periods during the COVID-19 pandemic. There is a clearly articulated rationale for this study and it provides a novel contribution, which extends existing literature. The methods are appropriate for addressing the research questions and the response rates were good for a two wave study. Utilising measures with clearly defined cut off criteria enables the authors to articulate the impact on mental health in a robust manner. Utilising a two-wave study also enables the authors to demonstrate the impacts over time.

The analyses are robust, however, more sophisticated analyses are available to demonstrate these findings, such as multilevel modelling, with variables nested within time. This would have provided some additional robustness to the analyses. However, as I have said, the analyses are robust and appropriate for addressing the research question.

The discussion is comprehensive and provides a plausible explanation for the findings, acknowledging the complexities of the different way employees were working in the context of the pandemic and multiple lockdowns. There are clear links to wider implications around the impacts and benefits from work and what can be learned from the pandemic can apply to current working practices and policies.

Overall, this was a comprehensive and thorough paper which was informative and a pleasure to read and learn from.

I do have a couple of very small amendments which I recommend, listed below:

*PDF page 11 - Other studies, focused on healthcare workers, have shown a positive relationship between psychological detachment from work, job stress and workload[13,44] – Please clarify this statement. The ‘positive relationship’ suggests that as psychological detachment increases, so does job stress and workload, Did you mean positively buffers against the impact of?

*PDF page 19 – Something missing from this statement – is it 7.1%?: “At T2, we observed 7.1 (N=169) to report low psychological detachment from work.”

Reviewer #2: This submission presents The Wellbeing of the Workforce (WoW) study, a prospective longitudinal cohort study, which conducted over two waves of data collection corresponding with the first and third national COVID-19 lockdowns in the UK. By examining the levels of psychological detachment from work during the initial lockdown and its impact one year later. The research offers valuable insights into how detachment can influence mental health outcomes in unprecedented times. Overall, I think this is a very interesting and well researched piece of work. I have some questions about the research, which if addressed sufficiently, would cement the study's rigor and original insight this works brings. I list these below:

Introduction

1. Report on the level of detachment and its impact on well-being that is already know from the literature. For example, in the introduction when this "well established" literature is mentioned. Then in the Discussion (see point 16.) what your study shows in comparison to that and if under these unprecedented times the amount of detachment appeared different, were relationships on key outcomes stronger etc.

2. More information about the social model that you used. So more about what is it, what does to comprise of, and how did it inform your research.

Method and Results

3. I'd like to know more about which variables were non-normal and whether not bootstrapping and/or alternative analysis approaches were considered. Given the extreme of the situation, and variables being not 'normal', it might be reasonable that some of these variables be skewed and alternative statistical tests be selected. Adjusting the data to fit normal models may not be accepted as appropriate here. However, some detail on this in the analysis section to illustrate the rationale would be good (and/or in the Discussion).

4. Report the power of your results from the data you have.

5. Completion rate was reported as 100% for T1 and T2. Later, you mention "Completion rate was the ratio of the number of people who finished each survey divided by those who completed the first page of the survey." Why was this decision made rather than completion rate indicting those fully completed each survey, for example. I suggest clarifying what you mean by completion rate and why you have taken this approach (with references)

6. Early you stated you replaced missed data with means as it was less than 5%. This may need to be clarified as representing "of those that responded" as around half of your sample from T1 did not provide data in T2 and could also be considered missing.

7. Related to point above, if you are using all the data from T1 in analysis of the smaller proportion (around half) in T2, I take it you are not replacing the missing respondents data? In which case, showing the sample characteristics at T1 is important to add. Changes may have occurred for example, which may have implications you wish to highlight in the discussion (e.g. job changes).

8. It would be helpful to understand sample characteristics at T1 the sample used in T2 that was matched for the longitudinal. Namely, to see if anything had changed. Perhaps add another column to table 1 to show the T2 sample at T1.

9. Provide rationale for why "participants one standard deviation below the mean was used to provide a threshold to quantify low psychological detachment from work" and any references for others that also used this cut-off or suggest its use. As well as why it was necessary to create a dichotomous variable for this construct. Discussion of the potential limitations of this approach will need to be presented in the Discussion section as there is literature suggesting this may not be appropriate.

10. Is there a missing % here in the line: One-fifth of the sample (21.4%; N=337), at T1, reported low psychological detachment from work. At T2, we observed 7.1 (N=169) to report low psychological detachment from work." Is this 7.1%?

11. What was the amount of detachment of the sample used in T2 at time point 1? Were there any differences there?

12. Were there differences in any of the other characteristics that you gathered data on? E.g full-time workers, key/essential workers, etc?

13. Covariates in regression did not include other factors examined in previous section, some of which showed significant differences between work status groups, for example. There are also other characteristics of the groups where data was collected but this has not been included in the inferential analyses. Adding the reasons why these were not added in this results section would be helpful (or if these have not been considered, consider analysing them). I see a comment is mentioned in the Discussion but I think providing the descriptives for these on your outcomes of interest would be helpful in the Results section where you can also show the insufficient data to perform the further analyses (e.g. Essential/key workers) as mentioned in point 12.

Discussion

14. Minor point, it may get confusing talking about COVID waves and waves of data collection. Consider using alternative phrasing for one.

15. Link to the stressor-detachment model is good to see. However, you discuss home working, which you did not analyse. More direct links from your findings and the model are needed and associated implications from them.

16. Add details of how detachment levels in your research compare to pre-pandemic research and what may be drawn from that.

17.Add a comment about the generalisability of your research. Perhaps also to our 'new ways of working'.

6. PLOS authors have the option to publish the peer review history of their article (what does this mean?). If published, this will include your full peer review and any attached files.

Reviewer #1: **Yes: **Dr Iain Wilson, Senior Lecturer in Social Sciences (Learning and Teaching)

Reviewer #2: No

---

## [Author Response · Author response to Decision Letter 0]

15 Jul 2024

Dear Editor,

Thank you for the opportunity to revise and resubmit the following manuscript to be considered for publication in PLOS ONE:

PONE-D-23-17342

Psychological detachment from work predicts mental wellbeing of working-age adults: findings from the ‘Wellbeing of the Workforce’ (WoW) prospective longitudinal cohort study

We have endeavoured to provide a point-by-point response and have uploaded the amended manuscript.

Editorial requests

We believe our manuscript meets the PLOS ONE style requirements. In addition, we have registered the protocol: https://doi.org/10.17605/OSF.IO/WA9B3

2. Open data requirement

We have submitted our raw data to the University of Nottingham Research Data Repository (doi: pending).

3. Data availability statement

The data underlying the results presented in the study are available from The Nottingham Research Data Management Repository (doi: pending).

4. Please review your reference list to ensure that it is complete and correct. 

References have been checked and are correct.

This reference (original ref 44) was deleted:

Viltea LS, Rodríguez-Carvajalb R, Hervás G. The role of emotion regulation strategies on healthcare workers’ mental health during the COVID-19. Ansiedad y Estrés. 2022;28(3):186-193. doi:10.5093/anyes2022a22.

Several new references have been added:

Eysenbach G. Improving the quality of Web surveys: the Checklist for Reporting Results of Internet E-Surveys (CHERRIES). J Med Internet Res. 2004 Sep 29;6(3):e34. doi: 10.2196/jmir.6.3.e34. Erratum in: doi:10.2196/jmir.2042.

Trógolo, M.A., Moretti, L.S. & Medrano, L.A. A nationwide cross-sectional study of workers’ mental health during the COVID-19 pandemic: Impact of changes in working conditions, financial hardships, psychological detachment from work and work-family interface. BMC Psychol 10, 73 (2022). https://doi.org/10.1186/s40359-022-00783-y

Sagherian K, McNeely C, Cho H, Steege LM. Nurses’ Rest Breaks and Fatigue: The Roles of Psychological Detachment and Workload. Western Journal of Nursing Research; 2023; 45; 10: 885-893.

Jiang W, Wang Y, Zhang J, Song D, Pu C, Shan C. The Impact of the Workload and Traumatic Stress on the Presenteeism of Midwives: The Mediating Effect of Psychological Detachment. Journal of Nursing Management, 2023: Article ID 1686151. https://doi.org/10.1155/2023/1686151

Hair Jr., J.F. and Fávero, L.P. (2019), "Multilevel modeling for longitudinal data: concepts and applications", RAUSP Management Journal, Vol. 54 No. 4, pp. 459-489. https://doi.org/10.1108/RAUSP-04-2019-0059

Faul, F., Erdfelder, E., Lang, A.-G., & Buchner, A. (2007). G*Power 3: A flexible statistical power analysis program for the social, behavioral, and biomedical sciences. Behavior Research Methods, 39, 175-191

Sandoval-Reyes, J., Restrepo-Castro, J. C., & Duque-Oliva, J. (2021). Work intensification and psychological detachment: The mediating role of job resources in health service workers. International Journal of Environmental Research and Public Health, 18(22), 12228.

Headrick, L., Newman, D.A., Park, Y.A. et al. Recovery Experiences for Work and Health Outcomes: A Meta-Analysis and Recovery-Engagement-Exhaustion Model. J Bus Psychol 38, 821–864 (2023). https://doi.org/10.1007/s10869-022-09821-3

Response to reviewers is within the next section.

Comments from Reviewer 1

Overall, this has been a comprehensive paper, covering key issues around the lockdown time periods during the COVID-19 pandemic. There is a clearly articulated rationale for this study, and it provides a novel contribution, which extends existing literature. The methods are appropriate for addressing the research questions and the response rates were good for a two-wave study. Utilising measures with clearly defined cut off criteria enables the authors to articulate the impact on mental health in a robust manner. Utilising a two-wave study also enables the authors to demonstrate the impacts over time.

Thank you for these positive comments.

The analyses are robust, however, more sophisticated analyses are available to demonstrate these findings, such as multilevel modelling, with variables nested within time. This would have provided some additional robustness to the analyses. However, as I have said, the analyses are robust and appropriate for addressing the research question.

We thank the reviewer for their reflections regarding the current robust analysis utilising regression analysis. We have highlighted within the limitations section some key reflections regarding the use of a regression analysis as opposed to a multi-level model. In particular, the potential of risk of standard errors being under-estimated (Hair & Favero, 2019). 

The discussion is comprehensive and provides a plausible explanation for the findings, acknowledging the complexities of the different way employees were working in the context of the pandemic and multiple lockdowns. There are clear links to wider implications around the impacts and benefits from work and what can be learned from the pandemic can apply to current working practices and policies. Overall, this was a comprehensive and thorough paper which was informative and a pleasure to read and learn from.

Thank you for these positive comments.

*PDF page 11 - Other studies, focused on healthcare workers, have shown a positive relationship between psychological detachment from work, job stress and workload[13,44] – Please clarify this statement. The ‘positive relationship’ suggests that as psychological detachment increases, so does job stress and workload, Did you mean positively buffers against the impact of?

We have re-written this paragraph, deleted one reference and used 3 new references. The relationship is now better described.

“Other studies, focused on healthcare workers, have shown that psychological detachment from work is positively associated with mental health (Trógolo et al, 2022), and is a mediator of within-shift work recovery (in the context of manageable workloads) (Sagherian et al, 2023), workload, traumatic stress and work presenteeism (Jiang et al, 2023)”.

*PDF page 19 – Something missing from this statement – is it 7.1%?: “At T2, we observed 7.1 (N=169) to report low psychological detachment from work.”

Thank you for this observation – we have added the missing %.

Comments from Reviewer 2

This submission presents The Wellbeing of the Workforce (WoW) study, a prospective longitudinal cohort study, which conducted over two waves of data collection corresponding with the first and third national COVID-19 lockdowns in the UK. By examining the levels of psychological detachment from work during the initial lockdown and its impact one year later. The research offers valuable insights into how detachment can influence mental health outcomes in unprecedented times. Overall, I think this is a very interesting and well researched piece of work. 

Thank you for the positive comments.

1. Report on the level of detachment and its impact on well-being that is already know from the literature. For example, in the introduction when this "well established" literature is mentioned. Then in the Discussion (see point 16.) what your study shows in comparison to that and if under these unprecedented times the amount of detachment appeared different, were relationships on key outcomes stronger etc.

We have integrated further discussion within the introduction to address the established literature on detachment and employee well-being. We've also tried to emulate this through the discussion. 

2. More information about the social model that you used. So more about what is it, what does to comprise of, and how did it inform your research.

We have provided further information regarding the theoretical model and discussion that informs this study and its postulated research questions, in the introduction. 

Method and Results

3. I'd like to know more about which variables were non-normal and whether not bootstrapping and/or alternative analysis approaches were considered. Given the extreme of the situation, and variables being not 'normal', it might be reasonable that some of these variables be skewed and alternative statistical tests be selected. Adjusting the data to fit normal models may not be accepted as appropriate here. However, some detail on this in the analysis section to illustrate the rationale would be good (and/or in the Discussion).

We have included as a supplementary material and overview of our normality testing across our variables. This provides an overview of the normality of our variables at the uni-variant level. However, for the purposes of our analysis we are particularly interested in multivariate levels of normality, which are our key concern regarding notable bias within the analysis conducted. In line with best practise, we have utilised an analysis with robust methods - drawing on bootstrapping as the optimal technique here. This technique allows us to estimate statistics that are reliable even when normality assumptions of these statistics are not met. 

4. Report the power of your results from the data you have.

Thank you for this suggestion. We have integrated in an analysis using G power conducting a post hoc power analysis. These results are integrated within our final write up. 

5. Completion rate was reported as 100% for T1 and T2. Later, you mention "Completion rate was the ratio of the number of people who finished each survey divided by those who completed the first page of the survey." Why was this decision made rather than completion rate indicting those fully completed each survey, for example. I suggest clarifying what you mean by completion rate and why you have taken this approach (with references)

A clear description of participation and completion rates is already provided under ‘Data Analysis’. We used the standard definitions of participation and completion rates that can be found in the Checklist for Reporting Results of Internet E-Surveys (CHERRIES). This reference has now been included.

Eysenbach G. Improving the quality of Web surveys: the Checklist for Reporting Results of Internet E-Surveys (CHERRIES). J Med Internet Res. 2004 Sep 29;6(3):e34. doi: 10.2196/jmir.6.3.e34. Erratum in: doi:10.2196/jmir.2042.

6. Early you stated you replaced missed data with means as it was less than 5%. This may need to be clarified as representing "of those that responded" as around half of your sample from T1 did not provide data in T2 and could also be considered missing.

Thank you for this observation, this has now been corrected.

7. Related to point above, if you are using all the data from T1 in analysis of the smaller proportion (around half) in T2, I take it you are not replacing the missing respondents’ data? In which case, showing the sample characteristics at T1 is important to add. Changes may have occurred for example, which may have implications you wish to highlight in the discussion (e.g. job changes).

We did not replace missing respondents’ data due to concerns regarding the introduction of a potential form of bias within the data source. 

However, to address this comment, we have integrated in a new table provided as a supplementary document that provides an overview of the sample size demographics at time one as compared to that observed at time 2. We have conducted a series of t tests to identify where -there may be significant changes in our observed demographics. 

8. It would be helpful to understand sample characteristics at T1 the sample used in T2 that was matched for the longitudinal. Namely, to see if anything had changed. Perhaps add another column to table 1 to show the T2 sample at T1.

We have integrated in a new table and analysis to demonstrate where there may have been notable changes from our time 1 to time 2 sample.

9. Provide rationale for why "participants one standard deviation below the mean was used to provide a threshold to quantify low psychological detachment from work" and any references for others that also used this cut-off or suggest its use. As well as why it was necessary to create a dichotomous variable for this construct. Discussion of the potential limitations of this approach will need to be presented in the Discussion section as there is literature suggesting this may not be appropriate.

We have provided reference support for this analytical decision and integrated key reflections in the discussion/ limitations section. 

10. Is there a missing % here in the line: One-fifth of the sample (21.4%; N=337), at T1, reported low psychological detachment from work. At T2, we observed 7.1 (N=169) to report low psychological detachment from work." Is this 7.1%?

Thank you for this observation, we have corrected the text.

11. What was the amount of detachment of the sample used in T2 at time point 1? Were there any differences there?

We have provided an additional analysis that highlights whether psychological detachment changed over time within our repeat measure sample and provided reflections when aggregated by work status. 

12. Were there differences in any of the other characteristics that you gathered data on? E.g full-time workers, key/essential workers, etc?

The central focus of this research study was to explore work status, rather than a wider selection of demographic and employment variables. However, we have conducted a regression analysis exploring whether any of the other demographic variables are statistically significant in predicting changes in psychological detachment. Only work status was a relevant and applicable variable. We have not detailed this analysis within the manuscript as would deviate from the scope of the research question. 

13. Covariates in regression did not include other factors examined in previous section, some of which showed significant differences between work status groups, for example. There are also other characteristics of the groups where data was collected but this has not been included in the inferential analyses. Adding the reasons why these were not added in this results section would be helpful (or if these have not been considered, consider analysing them). 

I see a comment is mentioned in the Discussion, but I think providing the descriptives for these on your outcomes of interest would be helpful in the Results section where you can also show the insufficient data to perform the further analyses (e.g. Essential/key workers) as mentioned in point 12.

Thank you for your observations. A key challenge was the smaller sample size available to us due to the repeat measure nature of our study. Therefore, we aimed to develop a regression model that was as parsimonious as possible in order to maintain statistical power. Therefore, only a small selection of study variables were entered into this multivariate analysis. This analytical consideration has been highlighted and discussed now within the discussion section. 

Discussion

14. Minor point, it may get confusing talking about COVID waves and waves of data collection. Consider using alternative phrasing for one.

We have talked about COVID surges, and lockdowns but do not refer to COVID waves.

All references to ‘waves’ relate to waves of data collection.

15. Link to the stressor-detachment model is good to see. However, you discuss home working, which you did not analyse. More direct links from your findings and the model are needed and associated implications from them.

Analysis of home working is not one of our study aims, so we have not analysed this. We have talked about remote working in the discussion as part of a critical reflection on what our findings may mean in the context of the COVID-19 pandemic and resulting changes for many people in ways of working (i.e., this is not a ‘finding’ per se, but relates to our interpretation and contextualising of our findings). 

We have referred to the model within the di

---

## [Decision Letter · Decision Letter 1]

11 Oct 2024

Psychological detachment from work predicts mental wellbeing of working-age adults: findings from the ‘Wellbeing of the Workforce’ (WoW) prospective longitudinal cohort study

PONE-D-23-17342R1

Dear Dr. Blake,

We’re pleased to inform you that your manuscript has been judged scientifically suitable for publication and will be formally accepted for publication once it meets all outstanding technical requirements.

Kind regards,

Laura Kelly, PhD

Division Editor

PLOS ONE

Additional Editor Comments (optional):

Reviewers' comments:

Reviewer's Responses to Questions

**Comments to the Author**

1. If the authors have adequately addressed your comments raised in a previous round of review and you feel that this manuscript is now acceptable for publication, you may indicate that here to bypass the “Comments to the Author” section, enter your conflict of interest statement in the “Confidential to Editor” section, and submit your "Accept" recommendation.

Reviewer #1: All comments have been addressed

Reviewer #2: All comments have been addressed

2. Is the manuscript technically sound, and do the data support the conclusions?

Reviewer #1: Yes

Reviewer #2: Yes

3. Has the statistical analysis been performed appropriately and rigorously? 

Reviewer #1: Yes

Reviewer #2: Yes

4. Have the authors made all data underlying the findings in their manuscript fully available?

Reviewer #1: Yes

Reviewer #2: Yes

5. Is the manuscript presented in an intelligible fashion and written in standard English?

Reviewer #1: Yes

Reviewer #2: Yes

6. Review Comments to the Author

Reviewer #1: Thank you for your considered response to my comments and the changes that you have made to the manuscript. My initial thoughts were that the manuscript was strong to start with and these changes only make it stronger.

I think that this is a novel piece of research with will contribute significantly to the literature. It has been conducted rigorously, which is demonstrated through the extensive literature and methods underpinning this work.

Overall, this is a strong manuscript.

Reviewer #2: Thank you for responding to the comments. A very interesting and valuable piece of work. Thank you for sharing it with us.

7. PLOS authors have the option to publish the peer review history of their article (what does this mean?). If published, this will include your full peer review and any attached files.

Reviewer #1: **Yes: **Dr Iain Wilson C.Psychol SFHEA, School of Social Sciences (Learning and Teaching), Nottingham Trent University

Reviewer #2: No

---

## [Editor Report · Acceptance letter]

30 Oct 2024

PONE-D-23-17342R1 

PLOS ONE

Dear Dr. Blake, 

I'm pleased to inform you that your manuscript has been deemed suitable for publication in PLOS ONE. Congratulations! Your manuscript is now being handed over to our production team.

Kind regards, 

on behalf of

Dr. Laura Hannah Kelly 

Staff Editor

PLOS ONE